# AskChart: Universal Chart Understanding through Textual Enhancement

## Abstract

Chart understanding tasks such as ChartQA and Chart-to-Text involve automatically extracting and interpreting key information from charts, enabling users to query or convert visual data into structured formats. State-of-the-art approaches primarily focus on visual cues from chart images, failing to *explicitly* incorporate rich textual information (e.g., data labels and axis labels) embedded within the charts. This textual information is vital for intuitive human comprehension and interpretation of charts. Moreover, existing models are often large and computationally intensive, limiting their practical applicability. In this paper, we introduce AskChart, a universal model that *explicitly* integrates both *textual* and *visual* cues from charts using a Mixture of Experts (MoE) architecture. AskChart facilitates the learning of enhanced visual-textual representations of charts for effectively handling multiple chart understanding tasks, while maintaining a smaller model size. To capture the synergy between visual and textual modalities, we curate a large-scale dataset named ChartBase with about 7.5M data samples, which helps align textual and visual information and facilitates the extraction of visual entities and text. To effectively train AskChart, we design a three-stage training strategy to align visual and textual modalities for learning robust visual-textual representations and optimizing the learning of the MoE layer. Extensive experiments across five datasets demonstrate the significant performance gains of AskChart in four chart understanding tasks. Remarkably, AskChart with 4.6B parameters outperforms state-of-the-art models with 13B parameters by **68.3%** in Open-ended ChartQA and **49.2%** in Chart-to-Text tasks, while achieving comparable performance in ChartQA and Chart-to-Table tasks.

## 1 Introduction

Charts are essential tools for data visualization, playing a crucial role in conveying complex data patterns in everyday applications (Wu et al., 2024). Chart understanding tasks, including chart question answering (ChartQA) (Hoque et al., 2022), Chart-to-Text (Kantharaj et al., 2022b), and Chart-to-Table translation (Liu et al., 2023), aim to automate the interpretation and extraction of key information from charts, allowing users to query or convert visual data into structured formats.

With the advancement of multimodal large language models (MLLMs), recent studies aim to automatically perform various chart understanding tasks (e.g., ChartQA and Chart-to-Text) by pretraining MLLMs on large-scale chart-related corpus (Masry et al., 2023; Han et al., 2023; Meng et al., 2024). For example, ChartAst (Meng et al., 2024) is trained on a large-scale instruction-following chart-related corpus based on Donut (Kim et al., 2022) and SPHINX (Lin et al., 2023) models, and demonstrates strong performance in ChartQA, Chart-to-Text and Chart-to-Table tasks.

Despite significant advancements, existing specialized MLLMs for chart understanding tasks predominantly rely on image-based representations, failing to *explicitly* leverage the rich textual information embedded in charts (Masry et al., 2023; Han et al., 2023; Meng et al., 2024). This limitation reduces their effectiveness, particularly in tasks requiring precise interpretation of textual content. For example, as shown in Figure 1(a), ChartAst (Meng et al., 2024) misrepresents key facts, such as the percentage of slices of the pie chart, due to inadequate integration of textual data.

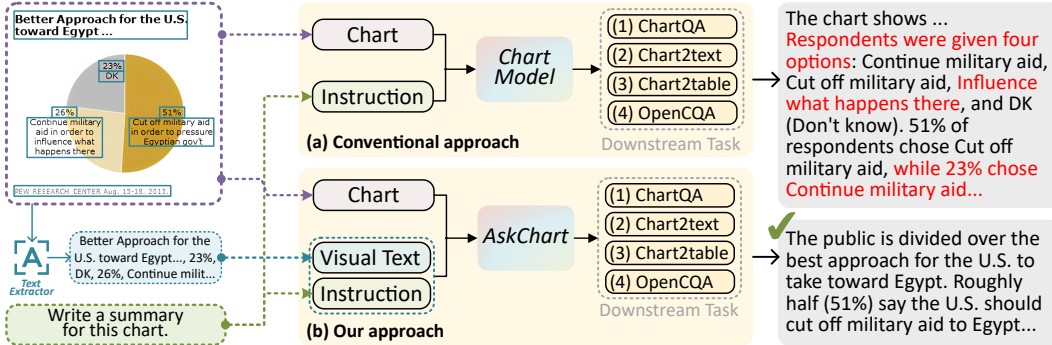

Figure 1: Comparison between the conventional approach (specialized MLLMs) and our proposed method (AskChart) for chart understanding tasks. Our approach explicitly integrates both visual and textual information from charts, resulting in better performance in chart understanding tasks.

**How do humans perform chart understanding?** Humans naturally "read" and "comprehend" charts by integrating both *textual* and *visual* information (Wu et al., 2024; Saket et al., 2019). When interpreting charts, people don't focus solely on visual elements like bars or lines. Instead, they actively incorporate textual cues such as axes and data labels to form a complete understanding of the data being presented. These textual elements provide essential context, clarifying relationships between variables, and resolving ambiguities in the graphical representation (Huang et al., 2024).

Inspired by this cognitive process, our **key idea** is to *explicitly* integrate textual information in chart understanding tasks, mimicking how humans interpret charts. To achieve this, as shown in Figure 1(b), our approach first employs a plug-in text extractor (e.g., OCR tools) to extract embedded textual information from the chart's visual elements and then aligns both visual and textual modalities to learn more effective joint representations. By explicitly combining both visual and textual cues, our approach could enable more accurate and comprehensive chart understanding, resulting in improved performance across various tasks such as Chart-to-Text and Open-ended ChartQA.

**Challenges.** Directly employing OCR tools to extract text from charts often results in errors such as misrecognition, incomplete extraction, or misalignment, particularly when dealing with complex chart structures. This presents the first challenge: *(C1: Alignment Challenge)* How to accurately align noisy OCR text with the corresponding visual components of the chart, enabling the model to learn meaningful joint representations and avoid misinterpretation? *(C2: Architectural Challenge)* How can we design a flexible and efficient architecture that can dynamically adapt to different chart types and tasks, effectively integrating visual and textual cues to optimize performance? *(C3: Dataset Challenge)* Existing datasets lack comprehensive training data that integrates both structural visual elements and textual information for chart understanding tasks.

**Our Methodology.** In response to these challenges, we introduce AskChart, a universal model that *explicitly* integrates both *textual* and *visual* cues from charts using a sparse Mixture of Experts (MoE) architecture to tackle multiple chart understanding tasks effectively. Specifically, AskChart utilizes a plug-in text extractor to extract textual information from charts, which is processed alongside user instructions via text encoders. In parallel, the visual encoder captures structural and visual chart information. The attention mechanism in LLMs integrates these components, while visual-textual alignment learning ensures the noisy extracted text is accurately aligned with its corresponding visual elements (addressing *C1*). To effectively handle diverse chart types and tasks without compromising on performance and efficiency, AskChart employs MoE layers, which allows for sparse computation, activating only the relevant experts and reducing unnecessary overhead by dynamically distributing tasks among specialized experts (addressing *C2*).

To address the third challenge (*C3*), we construct ChartBase, a large-scale dataset consisting of approximately 7.5 million samples that integrates both visual and textual elements from various chart-related tasks. ChartBase consists of three specialized datasets: *(a) the OCR-aware Data Prompt Dataset*: Aligns textual and visual information by featuring both single-turn and multi-turn instruction-following tasks, such as OpenCQA, Chart-to-Table, and chart summarization. *(b) Visual Prompt Dataset*: Comprising three types of chart question-answering tasks, i.e., reasoning, search, and data retrieval, where answers are visually highlighted using various prompt types (e.g.,

ellipses, bounding boxes, triangles) to enhance feature learning on chart images. *(c) the Chart-to-Table Instruction-Following Dataset*: Facilitates table and text extraction from charts.

**Contribution.** Our contributions can be summarized as follows:

(1) **New Methodology.** We propose AskChart, a lightweight model that explicitly integrates both textual and visual cues through MoE layers. We employ a three-stage training strategy with tailored pretraining objectives to enhance its performance across diverse chart understanding tasks.

(2) **New Dataset.** We introduce ChartBase, a large-scale dataset with approximately 7.5 million samples, comprising three specialized sub-datasets: the Visual Prompt Dataset, the OCR-aware Instruction-Following Dataset, and the Chart-to-Table Instruction-Following Dataset.

(3) **Extensive Experiments.** Our approach achieves new state-of-the-art performance across multiple benchmarks. AskChart outperforms larger models, such as those with 13B parameters, by 68.3% in Open-ended ChartQA and 49.2% in Chart-to-Text tasks, while delivering comparable results in ChartQA and Chart-to-Table tasks. We make both code and datasets publicly available at (`https://github.com/anonymousAskchart/AskChart`).

## 2 RELATED WORK

**Chart Understanding.** In chart understanding, key tasks have emerged, each focusing on interpreting and reasoning over chart data. ChartQA (Hoque et al., 2022; Xu et al., 2023) involves answering questions related to both the content and structure of charts, requiring models to extract insights from graphical elements. The Chart-to-Table (Liu et al., 2023) task converts visual chart data into structured tables for easier analysis, while Chart-to-Text (Kantharaj et al., 2022b) generates descriptive text from chart information. Complex tasks like Open-ended ChartQA (Open CQA) (Kantharaj et al., 2022a) demand higher-level reasoning beyond fact retrieval. Our AskChart is designed to handle these four core chart understanding tasks.

**MLLMs for Chart Understanding.** MLLMs like LLaVA (Liu et al., 2024b) and BLIP2 (Li et al., 2023a) have excelled in chart understanding tasks by leveraging abundant natural image datasets (Changpinyo et al., 2021; Lin et al., 2014; Liu et al., 2024b). However, high-quality pre-training datasets for charts are still underexplored. Existing methods like UniChart (Masry et al., 2023) expand task types but struggle with complex reasoning. Models like ChartLLaMA (Han et al., 2023), ChartAssistant (Meng et al., 2024), ChartGemma (Masry et al., 2024c), and ChartInstruct (Masry et al., 2024b) aim to address chart reasoning and editing tasks, while ChartMoE (Xu et al., 2024) improves multimodal input handling. However, open-ended tasks like OpenCQA (Kantharaj et al., 2022a) remain challenging. We propose AskChart with a visual-textual alignment pre-training approach that achieves state-of-the-art results in OpenCQA by better aligning visual chart structure with textual information of charts.

**Visual-Textual Alignment Learning.** Recent MLLMs (Zhang et al., 2023; Lin et al., 2024; Han et al., 2023) like LLaVA (Liu et al., 2024b) use single-turn conversations between humans and an assistant to briefly describe natural images. However, for charts, descriptions often include content that visual entities alone cannot capture (e.g., the semantic context of the chart) (Kantharaj et al., 2022b), which results in relatively noisy data for alignment tasks. Models like PresSTU (Kil et al., 2022), PaLI (Chen et al., 2022), and LLaVAR (Zhang et al., 2023) utilize noisy OCR-generated text as ground-truth prediction answers to enhance the model's text comprehension capabilities. Nevertheless, this noisy data remains insufficient for achieving robust alignment (Xu et al., 2020; Ren et al., 2016). LayoutLM (Xu et al., 2020) relies on object detection networks (Ren et al., 2016), which tend to underperform in charts that are rich in structural visual units, as they struggle to compute the patch-OCR loss to align vision and text. Similarly, ChartBERT (Xu et al., 2023), though using OCR-generated text, lacks the ability to effectively represent image and text information jointly. Limited approaches incorporate visual text as input for visual instruction fine-tuning. Our fundamental premise is to explicitly integrate visual-textual information with the user instruction, and then process them in parallel with the chart tokens through the training process of our AskChart.

## 3 ASKCHART MODEL

We will first present the architecture of AskChart (Section 3.1). We will then introduce the training objectives (Section 3.2) and finally elaborate on the training strategy (Section 3.3).

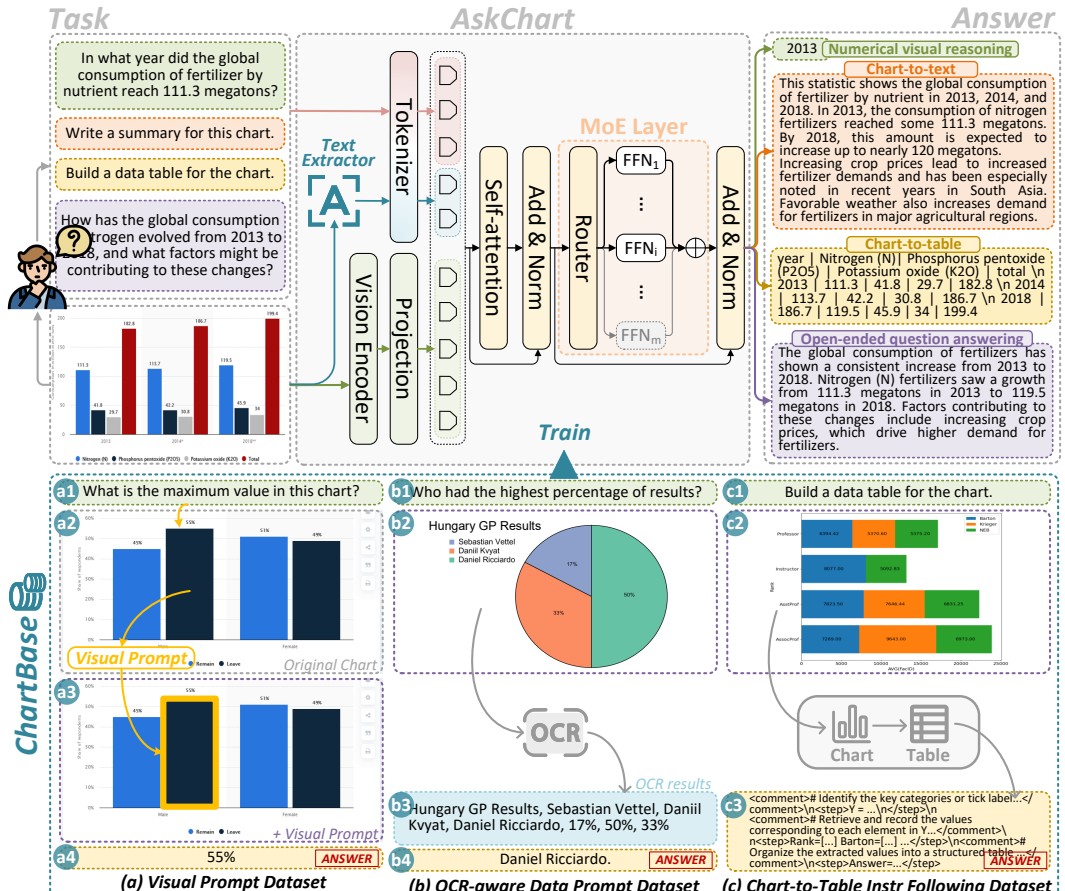

Figure 2: The framework of AskChart. The upper part shows the processing pipeline and AskChart structure while the lower part shows examples in ChartBase for pretraining. We newly curate three datasets: (a) Visual Prompt Dataset, (b) OCR-aware Data Prompt Dataset, and (c) Chart-to-Table Instruction Following Dataset. For ChartBase examples in lower part, blocks in green indicate *tasks* (a1, b1, c1); blocks with purple borders indicate *input charts* (a3, b2, c2); block in blue is the *OCR result* (b3); blocks in yellow indicate *answers* (a4, b4, c3).

### 3.1 ASKCHART ARCHITECTURE

**Overall Architecture.** As shown in Figure 2, the architecture of AskChart is designed to efficiently integrate both textual and visual information from charts. AskChart incorporates a text extraction module $\varphi(\cdot)$, which retrieves textual data from charts, alongside user instructions processed through a word embedding layer $g_t(\cdot)$. Simultaneously, a vision encoder $g_v(\cdot)$, captures the structural and visual elements. The extracted multimodal features are then aligned using a projection layer $proj(\cdot)$, and passed to an LLM, $f_\theta(\cdot)$. The LLM is enhanced with the MoE architecture, which dynamically allocates specialized experts to specific tokens. This design not only ensures efficiency and scalability but also enables the model to effectively manage the complex interactions between visual and textual modalities, all while maintaining a lightweight computational footprint.

To achieve a lightweight model, we adopt a tiny LLM (e.g., Phi) as a replacement for larger models like Vicuna (Chiang et al., 2023) and LLaMA (Touvron et al., 2023). Both the image encoder and LLM are built upon one of the recent state-of-the-art lightweight MLLMs, MoE-LLaVA (Lin et al., 2024). Given an input chart $\mathbf{X}_v$, the vision encoder processes the chart and generates a sequence of visual tokens. These tokens are then passed through a projection layer, which maps the visual tokens into language embedding tokens $\mathbf{H}_v$. Simultaneously, the text extractor processes the chart to extract visual text from the image, which is then combined with the user's instruction. Both the visual text $\mathbf{X}_o = \varphi(\mathbf{X}_v)$ and instructions $\mathbf{X}_t$ are passed through $g_t(\cdot)$ to generate visual-text sequence tokens $\mathbf{H}_o$ and instruction sequence tokens $\mathbf{H}_t$. Since the visual text is essentially textual information, we utilize the same text encoder for this task to simplify the process. Consequently, the

token sequences $\mathbf{H}_v$, $\mathbf{H}_t$, and $\mathbf{H}_o$ are concatenated and fed into the LLM, which uses MoE layers to replace the traditional feed-forward networks (FFNs). Each MoE block consists of a learnable router and multiple FFNs. The entire model workflow can be formally defined by the following equations:

$$\mathbf{H}_v = proj(g_v(\mathbf{X}_v)); \mathbf{H}_t = g_t(\mathbf{X}_t); \mathbf{H}_o = g_t(\mathbf{X}_o), \tag{1}$$

$$\mathcal{Y} = f_\theta([\mathbf{H}_v; \mathbf{H}_t; \mathbf{H}_o]), \tag{2}$$

where $\mathcal{Y}$ is the output answer.

**Text Extractor.** The text extractor is designed to accurately recognize task-agnostic visual text in charts with varying resolutions. Although some OCR-free vision encoders (Kim et al., 2022; Xu et al., 2020) trained on domain-specific data excel at understanding scene text, their generalization ability is limited, particularly when dealing with visual text in charts that vary in font size and style. Additionally, compared to some open-source OCR tools, these models often have a much larger number of parameters, making them difficult to deploy and fine-tune in resource-constrained environments. Therefore, we adopt a lightweight OCR tool, PaddleOCR (Pad), as the text extractor. Given a chart, PaddleOCR sequentially extracts the text by scanning from the top-left corner to the bottom-right corner of the image. The recognized visual text $\mathbf{X}_o$, which forms part of the LLM prompts used during both training and inference, is then concatenated with the user instruction $\mathbf{X}_t$.

## 3.2 TRAINING OBJECTIVES

We perform instruction-tuning of AskChart. Specifically, we train the LLM with MoE and the Vision Encoder in AskChart on the prediction tokens, using both the original (Lin et al., 2024) auto-regressive loss $\mathcal{L}_{reg}$ and an auxiliary loss $\mathcal{L}_{aux}$ (Fedus et al., 2021) which encourages the router to efficiently balance the load across multiple experts. The combined objective can be expressed as:

$$\mathcal{L} = \mathcal{L}_{reg} + \lambda \mathcal{L}_{aux}, \tag{3}$$

where $\lambda$ is a balancing factor that controls the contribution of the auxiliary loss $\mathcal{L}_{aux}$.

Given a sequence of length $L$, the auto-regressive loss of the target answers $\mathcal{Y}_a$ is defined as,

$$\mathcal{L}_{reg} = -\sum_{i=1}^{L} \log p_\theta \left( y_i \mid \mathbf{X}_v, \mathbf{X}_o, \mathbf{X}_{t,<i}, \mathcal{Y}_{a,<i} \right) , \tag{4}$$

where $\theta$ is the trainable parameters, $y_i$ is the current prediction token.

For $N$ experts, the auxiliary loss $\mathcal{L}_{aux}$ is computed as,

$$\mathcal{L}_{aux} = N \cdot \sum_{i=1}^{N} \mathcal{F}_i \cdot \mathcal{P}_i , \tag{5}$$

where $\mathcal{F}$ is the fraction of tokens processed by expert $i$, and $\mathcal{P}$ represents the portion of the router probability assigned to expert $i$, which can be defined as:

$$\mathcal{F}_i = \frac{1}{L} \sum_{i=1}^{L} \mathbf{1} \left\{ \arg\max p(x) = i \right\}; \ \mathcal{P}_i = \frac{1}{L} \sum_{i=1}^{L} p_i(x) . \tag{6}$$

## 3.3 TRAINING STRATEGY

To effectively train AskChart, we adopt a three-stage training strategy designed to align visual and textual modalities in charts, ensuring the model learns robust visual-textual representations. This strategy also fine-tunes the MoE layers to handle diverse chart understanding tasks efficiently. Throughout these stages, we employ multi-task tuning based on the ChartBase dataset (will be introduced in Section 4). Unlike existing MLLMs (Liu et al., 2024b; Lin et al., 2024; Meng et al., 2024), which typically freeze the vision encoder during training, we find that unfreezing the vision encoder across all stages significantly improves performance in chart understanding tasks.

Table 9 in the Appendix shows the tasks and datasets used across the different training stages.

**Stage I: Visual-Textual Alignment.** Effective chart understanding requires the model to establish a clear relationship between the chart's visual representation and its corresponding textual information. The goal of this stage is to accurately align noisy OCR-extracted text with the visual elements of the chart. To achieve this, we use Chart-to-Table translation as a pretraining task, similar to approaches used in ChartAst (Meng et al., 2024) and Matcha (Liu et al., 2022). The vision encoder and projection layer are trained to map image tokens into pseudo-text tokens. During this phase, we utilize relatively noisy chart-table pairs, where some of the underlying data tables are estimated based on the graphical marks (e.g., bars) as a percentage of the chart's plot area (Masry et al., 2023). Although this introduces some noise, we mitigate it with high-quality datasets during fine-tuning, effectively aiding the model in aligning charts with their corresponding tables.

**Stage II: Multi-task Instruction Tuning.** This stage aims to enable the model to generalize across various chart understanding tasks and diverse user instructions. As shown in Table 9, a key task is chart summarization, where the model generates summaries of chart content based on different user instructions, enhancing its ability to produce varying levels of detail. Specifically, Numerical and visual reasoning tasks go beyond the template-based reasoning seen in UniChart (Masry et al., 2023), by incorporating multi-turn conversations, covering sub-tasks like chart structural understanding, data retrieval, and mathematical reasoning. The open-ended ChartQA (Kantharaj et al., 2022a) task involves high-level questions requiring reasoning and explanatory answers. To address these, the model must comprehend visual text, demanding both perceptual and cognitive understanding. In contrast, low-level ChartQA tasks focus on specific goals such as reasoning, searching, and data retrieval. Each chart is marked with visual prompts to guide the model toward specific, highlighted areas of the image, improving task focus and accuracy.

**Stage III: Fine-tuning with Mixture of Experts.** To mitigate the learning difficulty associated with the sparse model architecture, we initialize the weights in the third stage using those from the second stage. When tokens are fed into the MoE layers, the router activates the top-$k$ experts to handle the tokens, and their outputs are combined using a weighted sum based on the router's weights. This mechanism helps distribute the computational load across multiple experts, improving the model's efficiency. In this stage, we fine-tune the model on tasks that are highly relevant to downstream tasks. Recognizing the challenges of translating charts to tables, we introduce a Chain-of-Thought (CoT)-based (Wei et al., 2022) translation task. This task requires the model to generate a step-by-step reasoning process (CoT) rather than producing a direct answer. By generating CoT answers, the model is encouraged to explicitly demonstrate its reasoning pathway, which leads to more accurate and interpretable results, particularly for complex Chart-to-Table translation tasks.

## 4 CHARTBASE DATASET

To enhance AskChart's chart understanding capabilities, we curate ChartBase, comprising three specialized datasets alongside existing work: (1) the Visual Prompt Dataset, (2) the OCR-aware Data Prompt Dataset, and (3) the Chart-to-Table Instruction-Following Dataset.

**ChartBase Overview.** Figure 2 illustrates examples from our ChartBase, and Appendix A provides a summary of the ChartBase statistics. Specifically, the Visual Prompt Dataset and OCR-aware Data Prompt Dataset cover 6 representative chart types: pie, common bar, stacked bar, grouped bar, common line, and grouped line charts. Among these types of charts, the common bar and common line both have only one category of data, while the stacked bar, grouped bar, and grouped line all have multiple categories of data. The Chart-to-Table Instruction Following Dataset additionally involves scatter plots. We transform all datasets, including datasets introduced by us and training sets of existing UniChart (Masry et al., 2023), ChatQA (Masry et al., 2022), OpenCQA (Kantharaj et al., 2022a), Chat-to-text (Kantharaj et al., 2022b) datasets, into an instruction-following format for pretraining. As shown in Appendix G.1, we design various instruction templates for random selection to increase language diversity. All the instruction-following datasets are used during the pretraining stages as illustrated in Table 9. Next, we will introduce the design consideration construction pipelines for each specialized dataset in ChartBase. For more details, please refer to Appendix G.

### 4.1 VISUAL PROMPT DATASET

Region understanding capabilities are crucial in chart understanding, as questions often target only particular elements, like individual bars in a bar chart. We also aim to strengthen the MLLM's nu-

merical visual reasoning to understand relationships among numerical values. Therefore, we develop and incorporate the Visual Prompt Dataset for second-stage pretraining, as shown in Figure 2(a).

**Construction.** Charts in ChatQA (Masry et al., 2022) are utilized as the foundation to construct the Visual Prompt Dataset. Firstly, we carefully design question templates (Appendix Table 14) to be used in question generation for four tasks: (1) reasoning, (2) extremum, (3) determining range, and (4) data retrieval. Subsequently, for each chart, we randomly select elements to generate questions and record their bounding box indices, thereby overlapping the visual prompt using ViP-LLaVA (Cai et al., 2024a). Charts unable to be visually prompted accurately by ViP-LLaVA, like involving correlation and distribution tasks, will be deemed unsuitable and consequently excluded. For diversity, we randomly select three types of visual prompts from a set of four (namely arrow, ellipsis, bounding box, and triangle) for each question, yielding 417,780 (Chart, Question, Answer) pairs ultimately. Figure 2-a2, a3 illustrates an example with the rectangle visual prompt.

### 4.2 OCR-AWARE DATA PROMPT DATASET

As mentioned, the weakness in text capture and utilization is a bottleneck limiting MLLMs' chart understanding capabilities. We aim to enhance MLLMs' such capabilities by providing richer and denser textual information aligned with the features in charts. Also, multi-turn question-answering examples are included to enable the model to better fit real-world scenarios. Therefore, we introduce the OCR-aware Data Prompt Dataset in the second-stage pretraining, as shown in Figure 2(b).

**Construction.** The OCR-aware Data Prompt Dataset includes two parts: single-turn and multi-turn instruction-following data, with each example comprising four essential elements: questions (Figure 2-b1), charts (Figure 2-b2), OCR results (Figure 2-b3), and answers (Figure 2-b4). For both single-turn and multi-turn examples, we employ PaddleOCR to extract textual information from the input charts to obtain OCR results. The single-turn instruction-following data is directly derived from UniChart (Masry et al., 2023) through format transformation, containing 6,791,230 examples. For multi-turn data, we utilize charts in UniChart accompanied by original tables, serving as the foundation for generation. First, we prompt ChatGPT (Ouyang et al., 2022) to identify and summarize the common question types in PlotQA (Methani et al., 2020) templates, which encompass three question-answering task categories: structural understanding, data retrieval, and mathematical reasoning. To enhance the effectiveness and accuracy of question and answer generation, we provide ChatGPT with sequenced original tables instead of charts. Then ChatGPT is prompted to synthetically generate two to three rounds of questions and answers, guided by identified question types (prompts in Appendix Table 15). Finally, we obtain 189,747 multi-turn examples.

### 4.3 CHART-TO-TABLE INSTRUCTION FOLLOWING DATASET

To improve AskChart's ability to comprehensively extract and understand information from charts, we propose COT based the Chart-to-Table Instruction Following Dataset for the third-stage fine-tuning, as shown by the example in Figure 2(c).

**Construction.** We construct a large amount of high-quality (chart, COT annotated table) pairs by converting tables into charts with COT ground-truth answers (see Appendix F). To this end, we first utilize widely used Text-to-SQL datasets, Spider (Yu et al., 2018) and BIRD (Li et al., 2024), which contain 1,020 and 1,460 tables on 138 and 37 domains respectively, as the base table. we first employ the automatic visualization system, DeepEye (Luo et al., 2018), to recommend good charts for these tables. Subsequently, we use Matplotlib to render the charts. Finally, we have a total of 61,472 (chart, table) pairs for forming our Chart-to-Table Dataset.

## 5 EXPERIMENTS

### 5.1 EXPERIMENTAL SETUP

**Datasets and Tasks.** We evaluate AskChart against state-of-the-art (SOTA) methods on four chart understanding tasks using various widely-used benchmarks. For ChartQA, we use the ChartQA benchmark (Masry et al., 2022), which focuses on visual and logical reasoning, where each question typically has a single word or numerical answer. This benchmark also includes the Chart-to-Table translation task, for which we follow the evaluation methodology from prior work. Additionally,

Table 1: Evaluation results on chart-related benchmarks. **Bold** indicates the best result, and underlined indicates the second-best result. The ChartQA task includes two sub-datasets (i.e., Aug. and Human), while other tasks similarly encompass one or more datasets.

| Categories | Model | Size | ChartQA | | | Open-ended ChartQA | Chart-to-Table | Chart-to-Text | |
|---|---|---|---|---|---|---|---|---|---|
| | | | Aug. | Human | Avg. | OpenCQA | ChartQA | Pew | Statista |
| General MLLMs | Blip2 (Li et al., 2023a) | 4B | 1.4 | 7.8 | 4.6 | 1.7 | - | 0.2 | 0.8 |
| | SPHINX (Lin et al., 2023) | 13B | 11.3 | 21.7 | 16.5 | 5.9 | 9.4 | 3.2 | 4.1 |
| | Qwen-VL (Bai et al., 2023) | 9.6B | 78.9 | 44.3 | 61.6 | 1.3 | - | 0.5 | 2.6 |
| Specialist Models | Pix2Struct (Lee et al., 2023) | 300M | 81.6 | 30.5 | 56.1 | 12.7 | 85.9 | 10.3 | 38.0 |
| | Chart-T5 (Zhou et al., 2023) | 400M | 74.4 | 31.8 | 53.1 | - | - | 9.1 | 37.5 |
| | Donut (Kim et al., 2022) | 260M | 78.1 | 29.8 | 54.0 | 13.1 | 87.4 | 7.2 | 38.2 |
| | Matcha (Liu et al., 2022) | 300M | 88.9 | 38.8 | 63.9 | 6.5 | 89.6 | 12.2 | 39.4 |
| | DePlot+Codex (Liu et al., 2023) | 1.3B+175B | 91.0 | 67.6 | 79.3 | - | 87.2 | - | - |
| | Unichart (Masry et al., 2023) | 260M | 88.6 | 43.9 | 66.3 | 14.8 | 91.1 | 12.5 | 38.1 |
| | OneChart+LLaVA1.6 (Chen et al., 2024) | 0.2B+34B | 85.3 | 49.1 | 67.2 | - | - | - | - |
| Chart MLLMs | ChartLLama (Han et al., 2023) | 13B | 90.4 | 48.9 | 69.7 | 4.7 | 90.0 | 14.2 | 40.7 |
| | ChartInstruct (Masry et al., 2024a) | 3B+7B | 93.8 | 50.2 | 72.0 | 14.8 | - | 12.8 | 40.1 |
| | TinyChart+PoT (Zhang et al., 2024) | 3B | 90.7 | 70.2 | 80.5 | 20.4 | 92.9 | 17.2 | - |
| | ChartAst-D (Meng et al., 2024) | 260M | 91.3 | 45.3 | 68.3 | 14.9 | 92.0 | 14.0 | 40.2 |
| | ChartAst-S (Meng et al., 2024) | 13B | 93.9 | 65.9 | 79.9 | 15.5 | 91.6 | 15.2 | 41.0 |
| | **AskChart (ours)** | 4.6B | 90.9 | 61.2 | 76.1 | **83.8** | 87.4 | **64.4** | **47.7** |

Table 2: Evaluation results on ChartInsights benchmark.

| Model | Size | Analysis | | | | Search | | | Query | | | Overall (%) |
|---|---|---|---|---|---|---|---|---|---|---|---|---|
| | | Reasoning | Anomaly | Distribution | Correlation | Range | Order | Filter | Retrieval | Extremum | Cluster | |
| VisCPM-Chat-v1.1 (Hu et al., 2023) | 10B | 28.4 | **46.1** | 33.3 | 51.9 | 23.0 | 6.4 | 25.1 | 15.8 | 32.0 | 29.6 | 26.2 |
| BLIP2 (Li et al., 2023a) | 11B | 24.8 | 23.4 | 25.0 | 15.1 | 25.3 | 20.2 | 39.8 | 27.8 | 30.3 | 30.1 | 28.3 |
| CogVLM-17B (Wang et al., 2024) | 17B | 20.3 | 23.1 | 43.6 | 29.6 | 37.7 | 10.8 | 9.1 | 37.9 | 56.6 | 26.7 | 29.4 |
| LLaVA1.5 (Liu et al., 2024b) | 13B | 32.4 | 6.3 | 30.9 | 23.1 | 21.7 | 32.7 | 35.6 | 32.6 | 35.8 | 43.5 | 32.2 |
| ChartAst-S (Meng et al., 2024) | 13B | 24.6 | 27.7 | 35.8 | 28.1 | 30.5 | 22.5 | 14.7 | 39.4 | 63.0 | 26.4 | 32.4 |
| MiniCPM-v2 (Hu et al., 2024) | 2.4B | 19.5 | 55.1 | 33.3 | 56.5 | 24.9 | 16.7 | 36.3 | 37.9 | 52.4 | 32.0 | 33.0 |
| mPLUG-Owl2 (Ye et al., 2023) | 7B | 31.0 | 27.0 | 29.4 | 35.3 | 28.4 | 22.5 | 40.3 | 30.9 | 41.1 | 27.3 | 33.3 |
| Qwen-VL (Bai et al., 2023) | 7B | 27.8 | 36.3 | 45.1 | 55.8 | 33.8 | 20.0 | 28.7 | 31.3 | 50.2 | 27.1 | 33.4 |
| ViP-LLaVA (Cai et al., 2024b) | 13B | 28.8 | 6.6 | 34.8 | 30.3 | 21.9 | **35.8** | **40.4** | 42.2 | 38.3 | 33.8 | 33.8 |
| LLaVA-NEXT (Liu et al., 2024a) | 13B | **30.6** | 7.4 | 26.5 | 38.0 | 29.5 | 33.3 | 23.4 | 53.5 | 59.8 | **52.3** | 38.5 |
| Sphinx (Lin et al., 2023) | 13B | 30.0 | 28.9 | 37.8 | 36.1 | 25.8 | 23.5 | 36.7 | 49.7 | **66.3** | 45.3 | 40.2 |
| **AskChart (ours)** | 4.6B | 28.6 | 21.5 | **50.5** | **58.7** | **59.5** | 10.4 | 27.3 | **71.2** | 52.8 | 31.5 | **42.7** |

we assess the model's performance in the chart summarization task using the Chart-to-Text benchmark (Kantharaj et al., 2022b). For Open-ended ChartQA, we evaluate using the OpenCQA benchmark (Kantharaj et al., 2022a), where questions require more explanatory and detailed answers.

**Evaluation Metrics.** We adopt evaluation metrics from prior studies (Masry et al., 2022). For ChartQA, we use relaxed accuracy (RA), allowing a 5% margin of error for numerical answers and exact matches for textual answers. For Chart-to-Table, we report RMS-F1 scores based on the DePlot framework (Liu et al., 2023). Both the Chart-to-Text task and OpenCQA are evaluated using BLEU scores (Post, 2018), consistent with previous works (Masry et al., 2023; Liu et al., 2022).

**Baselines.** We first selected several general-purpose MLLMs that excel at image understanding, such as Blip2 (Li et al., 2023a), SPHINX (Lin et al., 2023), and Qwen-VL (Bai et al., 2023), to evaluate their performance on chart-related tasks. Additionally, since Pix2Struct (Lee et al., 2023) and Donut (Kim et al., 2022) have shown impressive results in the domain of document understanding, we also considered specialist chart models like UniChart (Masry et al., 2023) and MatCha (Liu et al., 2022), which are based on these architectures and focus specifically on chart comprehension. Notably, Chart-T5 (Zhou et al., 2023) is an improved version of the versatile text-based language problem-solving model, T5 (Raffel et al., 2020). Some chart-specific MLLMs, built on popular vision-language models, can handle various chart-related tasks and achieve state-of-the-art performance, including models like ChartInstruct (Masry et al., 2024a), ChartLLaMa (Han et al., 2023) and ChartAst (Meng et al., 2024).

**Implementation Details.** AskChart is built on MoE-LLaVA (Lin et al., 2024), integrating SigLIP (Zhai et al., 2023) as the vision encoder and Phi-2 (Li et al., 2023b) as the language model. We trained all models using 8 A100 GPUs. Table 9 shows all datasets used for training. For Stage I, we trained the model for 1 epoch with a learning rate of 1e-3 and a batch size of 32 per GPU. For Stage II and Stage III, we fine-tuned the model for 1 and 6 epochs, respectively, with a learning rate of 2e-5 and a batch size of 16 per GPU. Please refer to Appendix D for more details.

## 5.2 MAIN RESULTS

Table 1 shows a comparison of AskChart with SOTA models across four chart understanding benchmarks. Remarkably, AskChart outperforms the current state-of-the-art methods by 68.3% and

Table 3: Ablation study on different prompts.

| Visual Prompt | Ocr-aware data prompt | ChartQA | | | Open-ended ChartQA | Chart-to-Table | Chart-to-Text | |
|---|---|---|---|---|---|---|---|---|
| | | aug. | human | avg. | OpenCQA | ChartQA | Pew | Statista |
| ✗ | ✗ | 75.5 | 44.9 | 60.2 | 63.1 | 63.9 | 55.2 | 55.1 |
| ✗ | ✓ | 83.8 | 50.1 | 67.0 | 79.3 | 81.3 | 57.2 | 58.0 |
| ✓ | ✗ | 76.6 | 46.1 | 61.4 | 63.4 | 62.6 | 50.9 | 55.1 |
| ✓ | ✓ | **84.6** | **50.9** | **67.8** | **79.3** | **81.5** | **60.6** | **62.8** |

49.2% (on the Pew sub-dataset), and 6.7% (on the Statista sub-dataset) in the open-ended ChartQA and chart-to-text tasks, respectively. This demonstrates that the lightweight AskChart (4.6B parameters) achieves competitive results on ChartQA and Chart-to-Table tasks, comparable to the performance of ChartAst-S (13B parameters). Notably, on the more computationally demanding ChartQA-human subset, which involves more challenging problems, AskChart surpasses the 13B ChartLLama (Han et al., 2023). We also observe that existing models struggle to effectively handle open-ended ChartQA, which requires generating explanatory answers by reasoning with the chart content. Due to its multitask training, AskChart performs joint visual and explicit text representation, and its MoE architecture enables a single token to be processed by different experts, with weighted outputs providing a more robust representation. This capability allows it to effectively address such complex tasks. Moreover, AskChart demonstrates significant advantages in tasks that demand both text recognition and generation. Unlike certain models, such as UniChart (Masry et al., 2023) and MatCha (Liu et al., 2022), which require fine-tuning for each downstream task to achieve optimal performance and often rely on separate models for different tasks, AskChart serves as a universal solution capable of addressing diverse requirements without task-specific fine-tuning.

Additionally, we conducted an error analysis based on chart types and question types (see Appendix B.1). From the accuracy distribution across different chart types, it is evident that the performance of AskChart is almost unaffected by the chart type, with comparable performance across various chart categories. To further analyze performance from the perspective of question types, we randomly selected 1,108 human-written questions. The model's performance was notably lower on data retrieval and compositional tasks that require multi-step reasoning, indicating that the vision encoder struggles with understanding chart values, while the large language model exhibits limitations in mathematical reasoning. These challenges primarily stem from the model's susceptibility to hallucinations in fine visual elements and its insufficient capacity for numerical representation.

### 5.3 FURTHER STUDY

The ChartInsights benchmark (Wu et al., 2024) evaluates multimodal models' capabilities in low-level chart analysis tasks, challenging them to not only recognize visual elements but also understand their underlying statistical and analytical significance. As shown in Table 2, AskChart demonstrates exceptional performance across various analytical tasks. Notably, it excels in the distribution and correlation tasks, achieving scores of 50% and 58.7%, the highest among all evaluated models. Furthermore, AskChart outperforms competitors in the range task with a leading score of 59.5%. Its performance in retrieval is also remarkable, achieving a score of 71%, significantly surpassing other models. Overall, AskChart attains an impressive total score of 42.7%, ranking first among all models. These results highlight the effectiveness of the OCR-aware data prompt strategy employed during pretraining, which has enabled AskChart to align textual and visual semantics effectively, particularly excelling in tasks requiring nuanced integration of both modalities.

### 5.4 ABLATION STUDY

**The Impact of Different Prompts.** To evaluate the influence of visual prompts and OCR-aware data prompts on model performance, we randomly sampled approximately 1M samples from the sub-datasets of each stage due to limited computational resources. We trained the model from scratch, and the results are shown in Table 3. The results indicate that visual prompts significantly enhance the model's performance on question-answering tasks (notably, we trained with only about 35% of the visual prompt dataset). This suggests that visual cues in charts help the model focus on the relevant areas associated with the questions.

**The Impact of Training Strategy.** To assess which alignment strategy more effectively aligns visual and textual information, we pre-trained the model in Stage I using two different tasks: Chart-to-

Text and Chart-to-Table. As shown in Table 4, the model trained with the Chart-to-Table alignment strategy consistently outperforms across multiple tasks. We attribute this to the fact that Chart-to-Table translation helps the model understand the underlying chart content rather than generating potentially irrelevant textual descriptions.

**The Impact of Number of Experts.** To evaluate the effect of the number of experts in the MoE layers on model performance, we conducted the following experiments. First, we varied the total number of experts while keeping the number of activated experts constant. As shown in Table 6, increasing the number of experts leads to improved performance across various tasks.

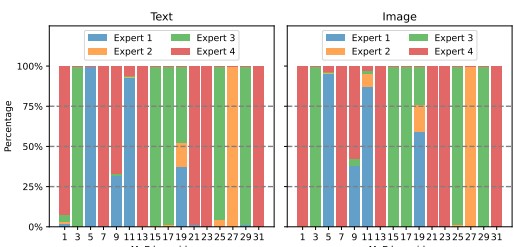

Figure 3: Modalities across different experts.

Table 4: Ablation study on training stage I.

| Task | ChartQA | Chart-to-Table | Chart-to-Text |
|---|---|---|---|
| w/ Chart2text | 67.0 | 80.2 | 59.1 |
| w/ Chart2table | **67.8** | **81.5** | **61.7** |

Table 5: Zero-shot study on multiple datasets.

| Model | ChartQA | Chart-to-Table | Chart-to-Text |
|---|---|---|---|
| | RealCQA | StructChart | ChartX |
| Unichart (Masry et al., 2023) | **38.0** | 1.6 | 6.8 |
| LLaVA1.5 (Liu et al., 2024b) | 30.0 | 7.5 | 0.45 |
| LLaVA-NEXT (Liu et al., 2024a) | 33.0 | 14.6 | 14.6 |
| ChartAst (Meng et al., 2024) | 11.0 | 14.3 | 12.8 |
| AskChart (ours) | 33.0 | **30.5** | **36.9** |

Table 6: The impact of the MoE layers.

| MoE Layers | ChartQA | Chart-to-Table | Chart-to-Text |
|---|---|---|---|
| w/o MoE | 35.9 | 59.1 | 31.2 |
| w/ MoE (#Experts=4) | **76.1** | **87.4** | **56.1** |

Table 7: The performance of top-$k$ experts

| Experts | ChartQA | Chart-to-Table | Chart-to-Text |
|---|---|---|---|
| 1 | 74.4 | 86.4 | 51.5 |
| 2 | **76.1** | **87.4** | **56.1** |

Furthermore, as illustrated in Figure 3, we examined the distribution of different modalities across the experts. Interestingly, the router distribution for both text and image tokens is similar, indicating that each expert is capable of processing both types of tokens. The weighted outputs from multiple experts contribute to stronger multimodal representations. Next, we varied the number of activated experts while keeping the total number of experts fixed. As presented in Table 7, activating 2 experts yields the best improvement in model performance. To balance computational efficiency and performance, we opted to set the number of activated experts to 2.

## 5.5 ZERO-SHOT STUDY

To evaluate the generalization capability of our model, we collected data from datasets that the model had never seen before for zero-shot experiments. Specifically, we conducted tests on several datasets, including RealCQA (Ahmed et al., 2023), StructChart (Xia et al., 2023), and ChartX (Xia et al., 2024), for the ChartQA, Chart-to-Table, and Chart-to-Text tasks, respectively. The evaluation metrics were consistent with those used for the corresponding tasks in the main results. As shown in Table 5, AskChart exhibited superior zero-shot performance across all tasks. In contrast, UniChart (Masry et al., 2023) performed poorly on both the Chart-to-Table and Chart-to-Text tasks, which we attribute to the limited language modeling capability of its text decoder. Even though ChartAst (Meng et al., 2024) utilizes a 13B parameter LLM, its generalization ability remains limited. AskChart, with only 4.6B parameters, demonstrated a clear advantage in ChartQA and text generation tasks. It suggests that the text-enhanced visual representation and robust MoE architecture contribute to the model's improved understanding of charts.

## 6 CONCLUSION

In this paper, we introduced AskChart, a lightweight chart understanding model that integrates both textual and visual cues using a Mixture of Experts architecture. By employing a three-stage training strategy with tailored pretraining objectives, AskChart demonstrates enhanced performance across diverse chart understanding tasks. We also presented ChartBase, a large-scale dataset with approximately 7.5M samples, featuring three specialized sub-datasets designed to improve the model's ability to comprehend and interpret chart data. Extensive experiments show that AskChart achieves state-of-the-art results, outperforming larger models in tasks such as Open-ended ChartQA and Chart-to-Text by 68.3% and 49.2%, respectively.

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

## A  STATISTICS OF CHARTBASE

Table 8: Statistics of ChartBase dataset.

| Subdatasets in ChartBase | #-Chart Types | #-Charts | #-Samples |
|---|---|---|---|
| Visual Prompt | 6 | 104,445 | 417,780 |
| OCR-aware Data Prompt | 6 | 505,037 | 6,980,977 |
| - Single-turn | 6 | 505,037 | 6,791,230 |
| - Multi-turn | 6 | 189,747 | 189,747 |
| Chart-to-Table Instr. Following | 7 | 61,472 | 61,472 |

Table 9: Tasks and datasets used for pretraining (Stage I and Stage II) and fine-tuning (Stage III). Our proposed dataset is denoted by "#". "*" indicates that only a subset of the dataset is used for the task. All datasets are accompanied by data prompts, except the visual prompt dataset.

| | Tasks | Datasets | #-Samples |
|---|---|---|---|
| Stage I | Chart-to-Table | #OCR-aware Data Prompt | 495K |
| Stage II | Chart Summarization | #OCR-aware Data Prompt | 481K |
| | Num & Vis Reasoning | #OCR-aware Data Prompt | 5.5M |
| | Open-ended ChartQA | #OCR-aware Data Prompt | 481K |
| | Low-level ChartQA | #Visual Prompt | 418K |
| Stage III | Chart-to-Text | Chart-to-Text (Kantharaj et al., 2022b) | 35K |
| | Open-ended ChartQA | OpenCQA (Kantharaj et al., 2022a) | 5K |
| | Chart-to-Table | *ChartQA (Masry et al., 2022) | 28K |
| | | #Chart-to-Table Instruction-Following | 61K |
| | Chart QA | *ChartQA (Masry et al., 2022) | 28K |

## B  ADDITIONAL RESULTS FROM EVALUATION

### B.1  ERROR ANALYSIS

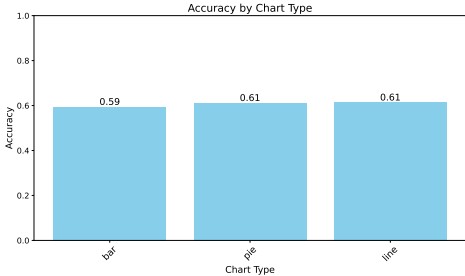

Figure 4: Results on the ChartQA Human test set by chart type.

Figure 5: Results on the ChartQA Human test set by question type.

Figure 4 presents the results across different chart types on the ChartQA-H benchmark. We randomly selected 1,108 human-written questions for this analysis. Figure 5 shows the performance breakdown by question type on ChartQA-H. The question types are as follows: (1) Data Retrieval: Questions focusing on directly extracting data information from the chart; (2) Visual Compositional: Tasks that involve identifying visual elements followed by reasoning to derive an answer; (3) Compositional: Multi-step reasoning or the combination of multiple pieces of information from the chart; (4) Visual: Questions that rely solely on the visual aspects of the chart to extract the answer, without requiring additional reasoning or composition.

### B.2  THE IMPACT OF MORE EXPERTS

We experimented with using more experts, as shown in Table 10. However, increasing the number of experts in the MOE architecture significantly inflates the model's parameter count, while the performance improvement is not proportional. As a result, we opted for a trade-off in the number of experts to leverage the advantages of the MOE framework fully. It is worth noting that extending the number of training steps might yield further performance gains.

Table 10: The impact of MoE experts numbers.

| #-Experts | ChartQA | Chart-to-Table | Chart-to-Text |
|:---------:|:-------:|:--------------:|:-------------:|
| 0 | 35.9 | 59.1 | 31.2 |
| 4 | 76.1 | 87.4 | 56.1 |
| 8 | 77.0 | 87.5 | 56.2 |

## C    LIMITATIONS

Although AskChart demonstrates competitive performance, hallucinations remain a challenge, particularly when reasoning about fine-grained visual elements within the chart. Future research could focus on enhancing the vision encoder's capabilities, potentially through strategies such as integrating multiple encoders or employing visual token merging techniques. Moreover, the inherent limitations of large language models in managing extended context lengths pose additional constraints. Input tokens exceeding a predefined length are truncated, potentially affecting training outcomes. Investigating methods to effectively support longer context lengths could be a promising direction for improving joint representations of visual and explicit textual information.

Regarding the experimental setup, it is important to note that most of the reported results are from a single run. Pretraining is computationally intensive and costly, particularly when multiple ablation setups are considered. We believe that the results would benefit from training over a greater number of steps.

## D    TRAINING DETAILS

Table 11: Training hyperparameters.

| Configurations | Stage I | Stage II | Stage III |
|:---|:---:|:---:|:---:|
| Experts | - | - | 4 |
| Top-$k$ | - | - | 2 |
| Deepspeed | Zero2 | Zero2 | Zero2 |
| Image resolution | | 384×384 | |
| Image encoder | | SigLip/384 | |
| Feature select layer | | -2 | |
| Image projector | | 2 Linear layers with GeLU | |
| Epoch | 1 | 1 | 6 |
| Learning rate | 1e-3 | 2e-5 | 2e-5 |
| Learning rate schdule | | Cosine | |
| Weight decay | | 0.0 | |
| Text max length | | 2048 | |
| Batch size per GPU | 32 | 16 | 16 |
| GPU | | 8 × A100-80G | |
| Precision | | Bf16 | |

We present the training hyperparameters for all stages, as shown in Table 11. We trained for 1 epoch in both of the first two stages, while in Stage III, due to the smaller dataset size, we trained for 6 epochs for appropriate total steps. The batch size was set to 256 in the first stage and 128 in the second and third stages. We utilized an image resolution of 384x384 across all three stages. Due to the excessive length of tokens extracted from the visual text, we encountered GPU out-of-memory issues in Stage III, even when using DeepSpeed's zero2_offload mode. To address this, we employed gradient accumulation.

## E    CHART UNDERSTANDING EXAMPLES

We below present examples for four involved chart understanding tasks: Chart-to-Text examples in Figure 6, Chart-to-Table examples in Figure 7, ChartQA examples in Figure 8, and OpenCQA examples in Figure 9.

Question: Write a summary for this chart

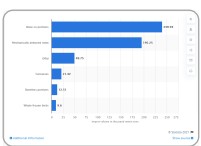

Ground Truth Answer:
This statistic shows the degree of urbanization in Myanmar, also known as Burma, from 2009 to 2019. Urbanization means the share of urban population in the total population of a country. In 2019, 30.85 percent of Myanmar's total population lived in urban areas and cities.

Predicted Answer:
This statistic shows the degree of urbanization in the Democratic Republic of the Congo from 2009 to 2019. Urbanization means the share of urban population in the total population of a country. In 2019, 30.85 percent of the total population of the Democratic Republic of the Congo lived in urban areas and cities.

Question: Provide a detailed description of the given image.

Ground Truth Answer:
This statistic shows the volume of frozen broiler meat imported to South Africa in 2016, by category. In that year, the volume of frozen offal imported to South Africa amounted to about 49.75 thousand metric tons.

Predicted Answer:
This statistic shows the import volume of poultry meat into the United States in 2018, by type. In that year, the United States imported approximately 239.59 thousand metric tons of bone-in portions of poultry meat.

Figure 6: Examples for Chart-to-Text tasks.

Question: Transform the visual information presented in the chart into an accurate tabular representation.

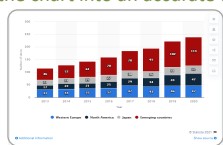

Ground Truth Answer:
Year | Western Europe | North America | Japan | Emerging countries\n2020 | 47 | 47 | 32 | 113\n2019 | 46 | 43 | 31 | 102\n2018 | 43 | 34 | 26 | 91\n2017 | 47 | 29 | 30 | 78\n2016 | 37 | 25 | 27 | 70\n2015 | 35 | 21 | 25 | 61\n2014 | 33 | 22 | 21 | 52\n2013 | 31 | 17 | 21 | 46.

Predicted Answer:
Year | Western Europe | North America | Japan | Emerging countries\n2020 | 47 | 47 | 32 | 113\n2019 | 46 | 43 | 31 | 102 \n2018 | 43 | 34 | 26 | 91 \n2017 | 47 | 29 | 30 | 78 \n2016 | 33 | 25 | 27 | 70 \n2015 | 35 | 21 | 21 | 61 \n2014 | 31 | 22 | 21 | 52 \n2013 | 17 | 11 | 17 | 46

Question: Transcribe the visual data points from the chart into a systematic table format.

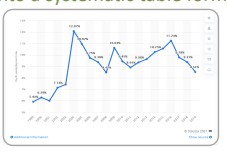

Ground Truth Answer:
Characteristic | Youth unemployment rate\n2019 | 8.51%\n2018 | 9.37%\n2017 | 9.78%\n2016 | 11.24%\n2015 | 10.56%\n2014 | 10.25%\n2013 | 9.64%\n2012 | 9.36%\n2011 | 8.91%\n2010 | 9.45%\n2009 | 10.63%\n2008 | 8.47%\n2007 | 9.38%\n2006 | 9.75%\n2005 | 10.97%\n2004 | 12.07%\n2003 | 7.43%\n2002 | 7.14%\n2001 | 6.01%\n2000 | 6.29%\n1999 | 5.92%

Predicted Answer:
Pred anwer: Characteristic | Youth unemployment rate\n2019 | 8.51%\n2018 | 9.37%\n2017 | 9.78%\n2016 | 11.24%\n2015 | 10.97%\n2014 | 10.25%\n2013 | 9.36%\n2012 | 8.91%\n2011 | 9.45%\n2010 | 8.91%\n2009 | 10.63%\n2008 | 9.37%\n2007 | 9.38%\n2006 | 9.75%\n2005 | 10.97%\n2004 | 12.07%\n2003 | 7.14%\n2002 | 6.97%\n2001 | 6.29%\n2000 | 6.29%\n1999 | 5.92%

Figure 7: Examples for Chart-to-Table tasks.

Question: How many companies were in Hungary's insurance market in 2013?

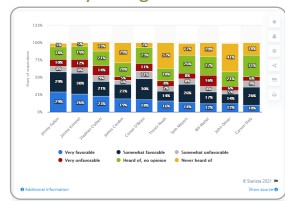

Ground Truth Answer: 36
Predicted Answer: 36

Question: Which late night host had the highest favorability ratings?

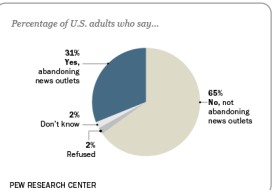

Ground Truth Answer: Jimmy Fallon
Predicted Answer: Jimmy Fallon

Question: What percentage of female students achieved a C/4 grade or higher in the United Kingdom in 2020?

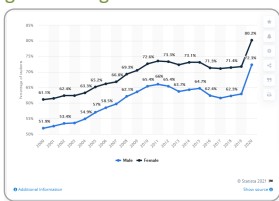

Ground Truth Answer: 80.2
Predicted Answer: 80.2

Question: What's the percentage of U.S adults who refused?

Ground Truth Answer: 2
Predicted Answer: 2

Figure 8: Examples for ChartQA tasks.

**Question:** How do married adults conceive their relationship satisfaction in comparison to adults living with a partner? Answer in a few sentences.

**Married adults have a more positive view of how things are going in their relationship**

*% of adults who are married or living with a partner saying things in their marriage/relationship are going ...*

| | Very well | Fairly well | Net |
|---|---|---|---|
| All married/Living with a partner | 54 | 39 | 94 |
| Married | 56 | 37 | 95 |
| Living with a partner | 41 | 47 | 88 |

*Source: Survey of U.S. adults conducted June 25-July 8, 2019.*
*"Marriage and Cohabitation in the U.S."*

**PEW RESEARCH CENTER**

**Ground Truth Answer:**
Married adults are more likely than those who are living with a partner to say things are going very well in their relationship (58 % vs. 41% They also express higher levels of satisfaction with specific aspects of their relationship, including the way household chores are divided between them and their spouse or partner how well their spouse or partner balances work and personal life, how well they and their spouse or partner communicate, and their spouse's or partner's approach to parenting.

**Predicted Answer:**
Married adults are more likely than those living with a partner to say things are going very well in their relationship. About six - in - ten married adults ( 58 % ) say this, compared with 41 % of those who are living with a partner.

**Question:** Describe the proportion of those supporting a way for undocumented migrants to gain legal status if certain requirements are met. Answer in a few sentences.

**Undocumented Immigrants in U.S. Who Meet Certain Requirements ...**

DK 3%
Should not be allowed to stay 25%
Should be allowed to stay in U.S. legally 71%
Apply for citizenship 42%
Apply for perm. res. 25%

*Survey conducted Oct. 15-20, 2014*

**PEW RESEARCH CENTER**

**Ground Truth Answer:**
When it comes to immigration policy, roughly seven - in - ten (71%) continue to support a way for undocumented immigrants to gain legal status if certain requirements are met.

**Predicted Answer:**
A substantial majority ( 71 % ) of the public says that undocumented immigrants who meet certain requirements should be allowed to stay in the U.S. legally.

Figure 9: Examples for OpenCQA tasks.

# F CoT CHART-TO-TABLE EXAMPLES

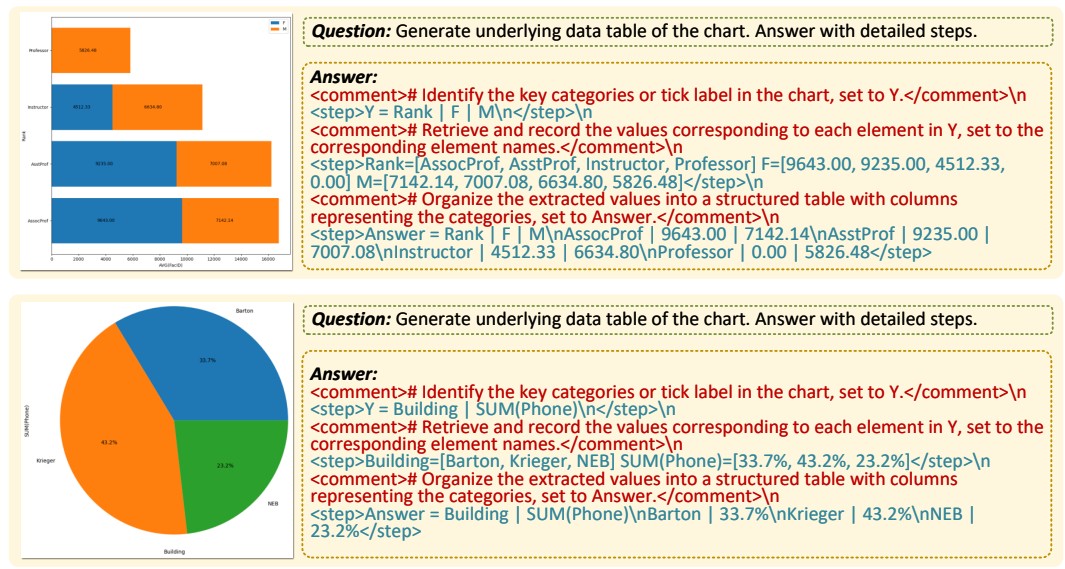

Figure 10: Two Examples of COT answers for Chart-to-Table instruction-following dataset.

Examples from the Chart-to-Table instruction-following dataset are shown in Figure 10. The CoT (Chain-of-Thought) answer involves a multi-step reasoning process, ultimately generating the required table.

# G DETAILS OF CHARTBASE

In this section, we report more detailed results of ChartBase.

## G.1 INSTRUCTION

We design various instruction templates to randomly select from for the chart2text and chart2table tasks, increasing expression diversity. Table 12 and Table 13 illustrate a portion of the instruction templates for chart2table and chart2text tasks, respectively.

Table 12: A portion of the instruction templates for the Chart-to-Table task.

| Instruction Template |
| --- |
| Extract and organize the data from the chart into a clear and concise table. |
| Create a detailed table reflecting the exact data points and categories shown in the chart. |
| Reconstruct the chart's data into a structured table, ensuring all elements are captured. |
| Translate the chart into a data table with precise values and labels as displayed. |
| Convert the charted information into a comprehensive table, including all relevant details. |
| Develop a tabular summary that encapsulates all the quantitative information from the chart. |
| Compile the data depicted in the chart into a well-organized table for easy interpretation. |
| Arrange the information contained within the chart into a methodical and detailed data table. |
| Replicate the chart's information accurately in table format, with corresponding data entries. |
| Catalog the chart data into a table, mirroring the exact figures and trends shown. |
| Transcribe the visual data points from the chart into a systematic table format. |

## G.2 VISUAL PROMPT

When creating a Visual Prompt dataset, we primarily follow two steps:

**STEP1: Make questions and get bounding boxes.** Step one is to identify the relevant elements and their bounding boxes based on the question. First, we generate the corresponding queries and answers according to the predefined question templates. For example, when generating a query about finding the maximum value in a bar chart, we construct the appropriate question and locate the maximum value in the chart. Since the dataset we are using includes the bounding box coordinates for each chart element, we can identify the element corresponding to the answer by referencing the question and find the bounding box coordinates for the bar representing the maximum value.

**STEP2: Generate Visual Prompts According to Bounding Boxes Automatically.** Step two is to automate the generation of the visual prompt using the bounding box. Here, we basically follow the rules in ViPLLaVA (Cai et al., 2024b). In our visual Prompt datasets, because we only have bounding boxes of each chart instead of pixel-level mask annotations, we only choose following visual prompt types: arrow, triangle, ellipsis, scribble, and bounding box. For the arrow, we make sure that the head of the arrow lies within $[(-\frac{W}{2}, -\frac{H}{2}),(\frac{W}{2}, \frac{H}{2})]$ space, where $W$,$H$ are the width and height of the image, respectively. For the triangle, We randomly select three points within the bounding box and connect them in sequence to form a triangle. For ellipse, the lengths along the semi-major and semi-minor axes are inherited from the bounding box size, where we enlarge the ellipse with a ratio between [1,1.5]. For scribble, we simulate human-like drawings using Bézier curves Farin (2014). This process begins by randomly selecting three points within the object mask, which serve as the anchors for the quadratic Bézier curve. The generated Bézier curve is then composited onto the image using the previously mentioned alpha blending technique to produce a merged image with the scribble serving as a visual prompt. Lastly, we use bounding box coordinates to draw relevant bounding boxes as visual prompts.

Figure 11 shows examples for each type of visual prompt.

## G.3 CHATGPT GENERATION PROMPT

We show the question templates in the Visual Prompt Dataset in Table 15.

Table 13: A portion of the instruction templates for the Chart-to-Text task.

| **Instruction Template for Brief Description** |
| --- |
| Describe the image concisely. |
| Provide a brief description of the given image. |
| Offer a succinct explanation of the picture presented. |
| Summarize the visual content of the image. |
| Give a short and clear explanation of the subsequent image. |
| Share a concise interpretation of the image provided. |
| Present a compact description of the photo's key features. |
| Relay a brief |
| clear account of the picture shown. |
| Render a clear and concise summary of the photo. |

| **Instruction Template for Detailed Description** |
| --- |
| Describe the following image in detail. |
| Provide a detailed description of the given image. |
| Give an elaborate explanation of the image you see. |
| Share a comprehensive rundown of the presented image. |
| Offer a thorough analysis of the image. |
| Explain the various aspects of the image before you. |
| Clarify the contents of the displayed image with great detail. |
| Characterize the image using a well-detailed description. |
| Break down the elements of the image in a detailed manner. |
| Walk through the important details of the image. |

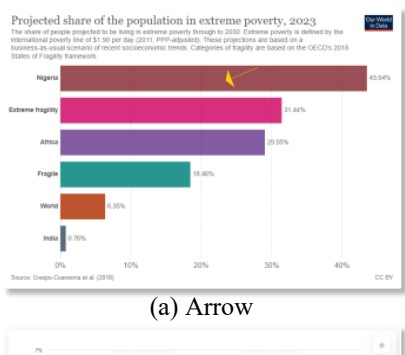

(a) Arrow

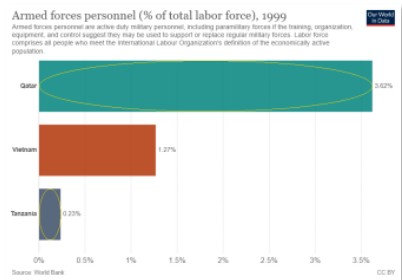

(b) Ellipsis

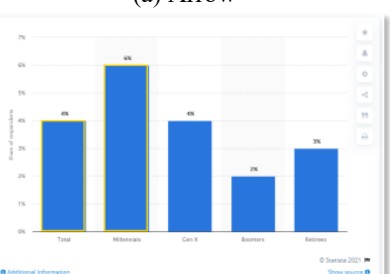

(c) Bounding Box

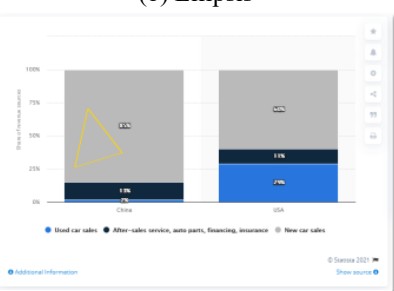

(d) Triangle

Figure 11: Four Types of Visual Prompt: Arrow, Ellipsis, Bounding Box, Triangle

Table 14: A portion of question templates in Visual Prompt Dataset

**Question Template for Reasoning**

What is the sum of {first_x_axis} and {second_x_axis} in this chart?

What is the difference of {first_x_axis} and {second_x_axis} in this chart? What is the mean value of {first_x_axis} and {second_x_axis} in this chart?

What is the total sum of all the elements in this chart?

What is the mean value of all the elements in this chart?

What is the sum of {first_x_axis} in {first_y_axis} and {second_x_axis} in {second_y_axis} in this chart?

What is the mean value of {first_x_axis} in {first_y_axis} and {second_x_axis} in {second_y_axis} in this chart?

What is the difference of {first_x_axis} in {first_y_axis} and {second_x_axis} in {second_y_axis} in this chart?

**Question Template for Extremum**

What is the maximum value in this bar chart?

What is the minimum value in this bar chart?

What is the maximum value in this line chart?

What is the minimum value in this line chart?

What is the maximum value in this pie chart?

What is the minimum value in this pie chart?

**Question Template for Determine Range**

What is the range of values in this bar chart?

What is the range of values in this line chart?

What is the range of values in this pie chart?

**Question Template for Data Retrieval**

How many bars are there in this bar chart?

How many pieces are there in this pie chart?

What is the value of {x_axis} in this chart?

What is the value of {x_axis} in {y_axis}?

Table 15: Prompt ChatGPT to generate multi-turn question-answer pairs based on underlying tables of charts to construct OCR-aware Data Prompt Dataset.

---

**🟢 Prompt for multi-turn question-answering generation**

You are an AI visual assistant that excels at chart figures. You are provided with a text description (chart summary) of a chart image and raw data values about the same chart. You don't have access to the actual image. Your task is to design question-answer pair(s) between a person (User) inquiring about the chart image and you (Assistant) responding to their questions.

---

Below are requirements for generating the question-answer pair(s):

- The answers should be a single word or phrase, and in a tone that a visual AI assistant is seeing the chart figure and answering the question.

- Ask diverse questions and give corresponding answers. Include questions asking about (1) various comparative aspects of chart image data, relationships between data points, changes over time or categories, and presence within specific ranges. (2) various numerical knowledge of chart data, including sums, differences, averages, medians, ratios, and statistical evaluations within the context of chart elements like legend labels and axis ticks or statistical measures like standard deviation, variance, and correlation and so on.

- The conversation should include at least 2-3 turns of questions and answers.

- Only include questions that have definite answers:(1) one can see in the chart figure that the question asks about and can answer confidently; (2) one can determine confidently from the chart figure that it is not in the chart figure. Do not ask any question that cannot be answered confidently.

- In addition, you are provided with some examples of question-answer pair(s) between a user and you(assistant).

[In context examples]

The chart description: [Description about chart figure]

The raw data: [Underlying data table]

---

