# OpenReview forum: "AskChart: Universal Chart Understanding through Textual Enhancement"
_ICLR.cc/2025/Conference — ICLR 2025 Conference Withdrawn Submission_

### Official Review · Reviewer_D1Yc · 2024-10-25

**Soundness:** 2
**Presentation:** 3
**Contribution:** 2
**Rating:** 5
**Confidence:** 4

**Summary:**

The paper presents AskChart, an innovative model designed for efficient chart understanding tasks by integrating textual and visual cues using a Mixture of Experts (MoE) architecture. AskChart addresses limitations in previous models by leveraging both visual and textual information in charts and providing a smaller yet highly effective model for multiple chart understanding tasks. The model also introduces ChartBase, a large-scale dataset that aligns textual and visual information to improve model performance.

**Strengths:**

1. The paper provides a coherent story, particularly in the Introduction section.

2. The authors have created a substantial amount of instruction-tuning data.

3. The three-stage training strategy introduced by the authors effectively aligns textual and visual information and optimizes MoE.

**Weaknesses:**

1. The use of MoE appears primarily aimed at reducing model parameters and computational overhead without a chart-specific design rationale. It would be helpful if MoE were more explicitly tailored for chart-specific tasks.

2. In practical deployment, it seems the model might first utilize OCR to extract text from charts and then combine this with instruction prompts to generate answers, similar to prior attempts. This approach may lack novelty.

3. The contribution is limited. The primary contribution seems to lie in the dataset aspect, yet the anonymized link provided contains no accessible data, leaving reviewers unable to evaluate the dataset itself.

4. The experimental comparison lacks completeness. Comparisons with Specialist Models like Deplot, General MLLMs such as LLaVA and mPLUG, and Chart MLLMs like TinyChart, OneChart, and Chartgemma are absent. Though the model has fewer than 7B parameters, the performance improvements over smaller models like TinyChart are marginal.

**Questions:**

1. There is a typo on line 282.

2. Why was a noisy OCR alignment used for the first alignment stage? Cleaner paired data is available in datasets such as MMC, ChartX, and ChartBench, which provide meta tables.

3. The authors introduce Chain-of-Thought (CoT) training in the third stage but do not explain how CoT intermediate results are validated. Did the authors provide instructions for the decomposition steps?

4. Given that balanced loss is used in the MoE architecture and each expert starts with the same initial parameters, why do the experts’ proportions differ so significantly in Fig. 3?

5. In Table 7, baseline results are very low. Did this baseline undergo the three-stage tuning used for the other experiments?

6. There is a significant difference in BLEU scores. In OpenCQA, for instance, other models score below 20, while AskChart reaches 83.8. Such a high score implies extreme lexical similarity with ground truth, suggesting potential overfitting. While the BLEU score is used for consistency with prior work, would LLM evaluators provide a more accurate metric?

7. AskChart’s generalization capabilities should be explored in benchmarks like SEED or MME. Does the training affect the original ability?

8. Table 2 and Table 4 present inconsistent results for AskChart. Please clarify.

9. Since OCR aids model performance, how does AskChart handle charts with limited data point annotations?

---

### Official Review · Reviewer_M78q · 2024-11-02

**Soundness:** 2
**Presentation:** 3
**Contribution:** 3
**Rating:** 5
**Confidence:** 3

**Summary:**

The work proposes AskChart, a lightweight Mixture of Experts model for chart understanding tasks. AskChart's novelty is that it explicitly aligns visual and textual information from charts. An LLM with a Mixture of Experts model generates the final answers based on those embeddings. The model is trained in a three-stage training procedure that requires explicit textual annotations. Therefore, the authors curate a new large-scale dataset called ChartBase, which includes multiple subsets of novel chart training data. For example, the Visual Prompt subset contains charts with visual cues on where the model should focus to generate its answer. The authors evaluate AskChart on four established chart understanding benchmarks and significantly improve the state-of-the-art performance for some of them, concluding that combining visual and textual cues is essential for chart understanding tasks.

**Strengths:**

- The paper's novelty is to explicitly incorporate textual information from charts inspired by the human cognitive system. The approach of the authors is promising for future research and generalizable to nearly every chart type.
- The authors present solid experimental results, aligning or improving the state-of-the-art of larger models.
- AskChart is a universal model that does not require task-specific fine-tuning.
- ChartBase provides a set of novel training data, explicitly incorporating visual cues and text alignment data.
- The overall idea of AskChart is easy to understand, and the presentation is done well. Figures help readers grasp the main concepts.
- The authors provide ablation studies for some design decisions, including the use of the Mixture of Experts model, and they examine the usefulness of the Visual and OCR-aware data prompt.

**Weaknesses:**

- The authors do not justify some of AskChart's design decisions. It would be interesting to examine whether a larger LLM improves AskChart's performance.
- Due to the many novel design decisions, it is hard to assess why AskChart shows performance leaps for some benchmarks. For example, although textual information is implicitly integrated, the model performs significantly worse than other models when not using a Mixture Of Experts model (see Table 7). It would be interesting to perform an extensive evaluation of the model components, especially the visual-textual alignment, without the impact of differing training data and further design decisions.
- There are some errors in writing, especially lines 459-461 and 369-471 need to be corrected.
- Some claims are not well-supported in the text: I did not find evidence for the claims "By generating CoT answers, the
model is encouraged to explicitly demonstrate its reasoning pathway, which leads to more accurate and interpretable results, particularly for complex Chart-to-Table translation tasks." (lines 308-310). Furthermore, Table 4 does not support the statement that "visual cues in charts help the model focus on the relevant areas associated with the questions." The inclusion of the visual prompts (rows 3 and 4 of Table 4) shows little to no improvements.

**Questions:**

- Could you provide details on the ablation studies reported in Table 4? Did you leave out the specific dataset for evaluations, or were some parts of the model architecture also cut out?
- I didn't fully understand Chapter 4.3, "Chart-To-Table Instruction Following Dataset." - The dataset is claimed to contain chain-of-thought ground truth answers. Please clarify or exemplify what you mean by chain-of-thought ground-truth tables.

---

### Official Review · Reviewer_Su6V · 2024-11-03

**Soundness:** 2
**Presentation:** 3
**Contribution:** 2
**Rating:** 5
**Confidence:** 2

**Summary:**

This paper's main contribution is AskChart, a lightweight model for chart understanding. Motivated by the importance of textual elements in visualizations, the authors propose an approach that integrates both textual and visual cues in charts. MoE layers are further added to enhance performance. A secondary contribution is ChartBase, a dataset for multi-task tuning.

**Strengths:**

+ The overall narrative is quite easy to follow. The authors gave strong motivations for why incorporating textual elements is important in chart understanding, presented their training procedures pretty clearly, and wrote clearly about their experimental results.
+ The performance of AskChart seems quite amazing on several tasks (e.g., OpenCQA), leading by a wide margin.

**Weaknesses:**

I want to preface this by saying that I am a relatively junior reviewer, so please take my comments with a grain of salt.
- My main complaint is that the benchmarking procedure might be unfair. The authors mentioned that two of the datasets they pretrained/tuned AskChart on were OpenCQA and Chart-To-Text, but then the model was also tested on OpenCQA and Chart-To-Text, and achieved 68.3% and 50% leads ahead of the second-best models. Perhaps the fact that the authors pretrained their model on the test set explains why there is a huge gap for those tasks.
- [related]: I think the paper could benefit from a more in-depth discussion of why for some tasks AskChart leads by a lot, while for some other tasks it is not in the top 2 or 3.
- Explicitly incorporating textual elements in chart understanding is not a new idea. I think the authors need to more explicitly acknowledge it. The way the authors wrote the paper sounds as if they delved into how humans read charts and came up with this new approach themselves (see paragraphs beginning with "How do humans perform chart understanding?") However, at least as early as 2007 (A system for understanding imaged infographics and its applications), people have tried improving chart understanding by considering text and visual elements. For a review, see From Pixels to Insights: A Survey on Automatic Chart Understanding in the Era of Large Foundation Models. The authors may be one of the first to take advantage of this fact in a MLLM setting, but they should at least recognize previous efforts that inspired them.

**Questions:**

1. In the contribution section of the introduction, the authors claimed ChartBase is a benchmark. But it seems more like a pretraining/tuning dataset to me the way it is used in the paper. Could the authors clarify this please?
2. Why are only two models compared against AskChart in Table 6 (instead of the wholistic comparison done previously)? Which version of ChartAst is the table referring to?

---

### Official Review · Reviewer_9cbi · 2024-11-05

**Soundness:** 1
**Presentation:** 3
**Contribution:** 3
**Rating:** 6
**Confidence:** 2

**Summary:**

The paper presents AskChart, a lightweight chart understanding model that integrates textual and visual cues from charts using a Mixture of Experts (MoE) architecture. The authors propose a three-stage training strategy and introduce a large-scale dataset, ChartBase, comprising approximately 7.5 million samples to enhance the model's ability to comprehend and interpret chart data. Experimental results demonstrate that AskChart achieves superior performance across multiple chart understanding tasks, particularly in open-ended ChartQA and Chart-to-Text tasks, where it significantly outperforms models with more parameters.

**Strengths:**

1. AskChart effectively integrates textual and visual information from charts through a MoE architecture, which not only keeps the model lightweight but also improves efficiency and scalability by dynamically assigning specific experts to handle particular tasks.

2. The authors have constructed a substantial dataset, ChartBase, which includes around 7.5 million samples. This dataset provides a rich resource for model training and evaluation and contributes significantly to the field of chart understanding.

3. AskChart shows good performance in various chart understanding tasks, especially in open-ended ChartQA and Chart-to-Text tasks, where it achieves significant performance improvements over models with a larger parameter count, demonstrating the effectiveness and superiority of their approach.

**Weaknesses:**

1. What is the impact of OCR tools on AskChart? AskChart uses the lightweight OCR tool PaddleOCR as a text extractor, which may result in performance loss. How significant is this loss for AskChart? It can be considered the effects of different OCR tools (OCR recognition accuracy) in this context.

2. In line 264, the paper states that unlocking the visual encoder at all stages improves the performance of the chart understanding task, but there is no experimental evidence or reasonable justification for the performance improvement provided.

3. It is noted that the paper introduces CoT reasoning in the third phase to address the challenge of converting charts into tables. However, more details are not provided, such as how the final answer from CoT is obtained.

4. The paper introduces $𝐿_{aux}$, but does not analyze the impact of its weight (λ).

5. The analysis of the number of MoE experts is insufficient (Tables 7 and 8).

**Questions:**

Why do the results remain the same despite the different number of experts in Tables 7 and 8?

---

### Official Review · Reviewer_ZVX2 · 2024-11-08

**Soundness:** 3
**Presentation:** 3
**Contribution:** 3
**Rating:** 6
**Confidence:** 4

**Summary:**

This study introduces AskChart, a model for chart understanding tasks like ChartQA and Chart-to-Text. Unlike existing approaches that mainly rely on visual cues, AskChart integrates both visual and textual information (e.g., data labels) from charts using a Mixture of Experts (MoE) architecture. The model is trained on a large-scale dataset called ChartBase (7.5M samples) to effectively align these modalities. AskChart uses a three-stage training strategy and significantly outperforms larger models, achieving notable gains in Open-ended ChartQA and Chart-to-Text tasks with only 4.6B parameters, compared to 13B-parameter models.

**Strengths:**

1. The chart question answering task addressed in this paper is highly applicable to real-world scenarios, underscoring its strong practical relevance and substantial research significance.
2. The primary advantage of the proposed framework lies in its lightweight design, which enhances its practicality and makes it well-suited for real-world applications.
3. The benchmark constructed in this paper is substantial in scale and comprehensively considers a wide range of aspects, which will be instrumental in advancing future research in the related domain.

**Weaknesses:**

1. The use of OCR to extract textual elements from charts appears to have been explored in prior works, and thus does not represent a novel perspective for addressing this task. Given this context, what would you consider to be the primary advantage of your approach compared to these existing methods? Please clarify how your work differentiates itself and advances beyond these earlier attempts.
2. The paper contains several minor grammatical and formatting issues that should be addressed. For instance, on line 282, a period is missing before "To achieve this." Additionally, in Table 7, there is an unexplained question mark next to "MoE." These errors, while small, detract from the overall clarity and polish of the manuscript.
3.  The proposed Askchart appears to lack significant algorithmic innovation, with the primary contributions focusing on dataset integration and further refinement. While the amount of work involved is substantial, the level of novelty in terms of algorithmic development is insufficient to meet the standard of methodological innovation expected in this field.
4. What are the limitations of AskChart? It seems that the authors have not provided an adequate discussion on this aspect. A thorough examination of the limitations would offer a more balanced perspective on the proposed approach and highlight potential avenues for future research.
5. I recommend that the authors provide an error analysis to enhance the comprehensiveness of the experimental section. Such an analysis would offer valuable insights into the model's shortcomings and help elucidate potential areas for improvement.

**Questions:**

1. The use of OCR to extract textual elements from charts appears to have been explored in prior works, and thus does not represent a novel perspective for addressing this task. Given this context, what would you consider to be the primary advantage of your approach compared to these existing methods? Please clarify how your work differentiates itself and advances beyond these earlier attempts.
2. The paper contains several minor grammatical and formatting issues that should be addressed. For instance, on line 282, a period is missing before "To achieve this." Additionally, in Table 7, there is an unexplained question mark next to "MoE." These errors, while small, detract from the overall clarity and polish of the manuscript.
3.  The proposed Askchart appears to lack significant algorithmic innovation, with the primary contributions focusing on dataset integration and further refinement. While the amount of work involved is substantial, the level of novelty in terms of algorithmic development is insufficient to meet the standard of methodological innovation expected in this field.
4. What are the limitations of AskChart? It seems that the authors have not provided an adequate discussion on this aspect. A thorough examination of the limitations would offer a more balanced perspective on the proposed approach and highlight potential avenues for future research.
5. I recommend that the authors provide an error analysis to enhance the comprehensiveness of the experimental section. Such an analysis would offer valuable insights into the model's shortcomings and help elucidate potential areas for improvement.

---

### Author Response · Authors · 2024-11-24

Dear reviewers,

Thank you again for your valuable feedback! This is a gentle reminder that the discussion period is nearing its conclusion, we hope you have taken the time to consider our responses to your review. The revised parts are marked as blue. If you have any additional questions or concerns, please let us know so we can resolve them before the discussion period concludes. If you feel our responses have satisfactorily addressed your concerns, it would be greatly appreciated if you could raise your score to show that the existing concerns have been addressed.

Thank you!

Regards,
Authors

---

### Note · Authors · 2025-01-22

I have read and agree with the venue's withdrawal policy on behalf of myself and my co-authors.